# CAST: Modeling Visual State Transitions for Consistent Video Retrieval

**Yanqing Liu** [1 2 *]   **Yingcheng Liu** [1 3]   **Fanghong Dong** [1]   **Budianto Budianto** [1]   **Cihang Xie** [2]   **Yan Jiao** [1]

## Abstract

As video content creation shifts toward long-form narratives, composing short clips into coherent storylines becomes increasingly important. However, prevailing retrieval formulations remain context-agnostic at inference time, prioritizing local semantic alignment while neglecting the state and identity consistency. To address this structural limitation, we formalize the task of **Consistent Video Retrieval (CVR)** and introduce a diagnostic benchmark spanning YouCook2, COIN, and CrossTask. We propose **CAST (Context-Aware State Transition)**, a lightweight, plug-and-play adapter compatible with diverse frozen vision-language embedding spaces. By predicting a state-conditioned residual update ($\Delta$) from visual history, CAST introduces an explicit inductive bias for latent state evolution. Extensive experiments show that CAST improves performance on YouCook2 and CrossTask, remains competitive on COIN, and consistently outperforms zero-shot baselines across diverse foundation backbones. Furthermore, CAST provides a useful reranking signal for black-box video generation candidates (e.g., from Veo), promoting more temporally coherent continuations.

## 1. Introduction

Video content creation is increasingly shifting from isolated clips to structured multi-step compositions, driven by recent advances in large-scale video generation models (Villegas et al., 2022; OpenAI, 2024; Team et al., 2025; Google DeepMind, 2024). A central challenge in this setting is *composition*: selecting and ordering short video segments to form a temporally coherent sequence. Procedural activities, such as cooking tutorials (e.g., "wash" $\rightarrow$ "cut" $\rightarrow$ "cook"), provide a concrete and widely studied instance of this problem, where each step requires retrieving the appropriate next segment from a candidate set. Yet existing retrieval approaches treat these selections independently, prioritizing local semantic alignment while often producing fragmented narratives.

This limitation is rooted in the design of standard video retrieval itself. To enable scalable indexing, prevailing formulations encode clips independently and therefore operate under a context-agnostic inference paradigm (Miech et al., 2019; Luo et al., 2021; Bain et al., 2021; Zhao et al., 2024; Wang et al., 2022). In practice, this design induces two recurrent forms of incoherence. The first is identity inconsistency: abrupt shifts in actor, environment, or visual style across consecutive clips. The second, particularly salient in procedural settings, is state inconsistency: violations of procedural causality. For example, as illustrated in Figure 1, given a context clip of a person preparing ingredients, a query for "slice the tomatoes" may retrieve a later step showing *already sliced* tomatoes being plated (State Error), or a clip of a different person cutting on a mismatched *wooden board* (Identity Error), rather than the correct continuation of the original actor. Thus, although current methods achieve strong semantic recall on standard video retrieval benchmarks such as MSR-VTT (Xu et al., 2016), these metrics often fail to capture the temporal consistency required for coherent storytelling. Crucially, this failure mode does not primarily arise from weak visual or textual representations; rather, it reflects the absence of an explicit inductive bias for modeling state evolution under contextual constraints.

To address this gap, we introduce **Consistent Video Retrieval (CVR)**. Moving beyond standard text-to-video retrieval, CVR recasts retrieval as a *context-aware* inference problem: given a sequence of preceding clips and a text instruction, the model must retrieve a target clip that is not only semantically aligned with the instruction, but also consistent with the preceding visual state and identity cues. To evaluate this setting rigorously, we establish a protocol derived from procedural datasets: YouCook2 (Zhou et al., 2018), COIN (Tang et al., 2019), and CrossTask (Zhukov

---

*This work was done while Yanqing was a research intern at Google. [1]Google, Sunnyvale, CA, USA [2]Department of Computer Science and Engineering, University of California, Santa Cruz, Santa Cruz, CA, USA [3]Massachusetts Institute of Technology, Cambridge, MA, USA. Correspondence to: Yanqing Liu <yliu858@ucsc.edu>, Yan Jiao <yanjiao@google.com>.

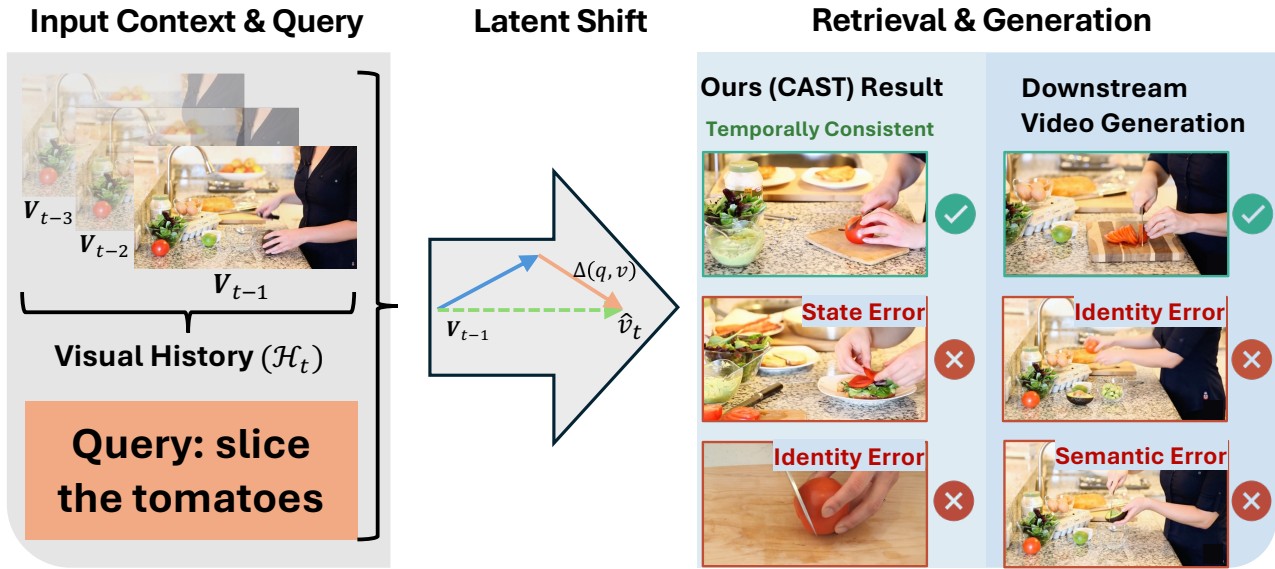

*Figure 1.* Given a context clip and instruction, standard retrieval often returns semantically relevant but temporally incoherent clips, yielding **State Errors** or **Identity Errors**. In contrast, **CAST** models the state transition ($\Delta$) to retrieve a causally plausible continuation and rerank generation candidates toward more coherent continuations.

et al., 2019). While our broader motivation is long-horizon visual coherence, we focus on procedural activities as a controlled testbed. Unlike open-ended movies, procedural activities exhibit strict temporal ordering constraints, where steps cannot be arbitrarily permuted (Alayrac et al., 2016; Zhou et al., 2018; Zhukov et al., 2019). These constraints often manifest as persistent changes in the underlying visual state (e.g., an egg cannot be un-cracked). Accordingly, unlike standard benchmarks dominated by semantically distinct distractors, our protocol constructs hard negatives that are either visually context-preserving or semantically related while differing in state or identity, making retrieval metrics explicitly sensitive to consistency errors.

Building on this formulation, we propose **CAST** (**C**ontext-**A**ware **S**tate **T**ransition), motivated by the observation that actions in procedural narratives are better understood as latent visual state transitions than as static semantic labels. CAST therefore models procedural progression as a sequence of state-conditioned transitions. In contrast to context-agnostic matching, CAST uses the text instruction to predict a **residual vector ($\Delta$)** in the embedding space. This residual updates the current state embedding, modifying procedural attributes while retaining identity-relevant information from the anchor state through the additive connection (He et al., 2016). Formally, this yields the retrieval target $\hat{v}_t \approx v_{t-1} + \Delta(v_{t-1}, q_t)$ (see §2.3 for full formulation), enabling the model to search for a causally plausible continuation rather than merely a semantic match. CAST is implemented as a lightweight trainable adapter (Houlsby et al., 2019; Hu et al., 2022; Gao et al., 2023) atop frozen foundation models, while remaining plug-and-play at infer-

ence time. Extensive experiments on our CVR benchmark show that this transition-based formulation consistently outperforms context-agnostic baselines, yielding stronger state discrimination while retaining identity cues and semantic recall. Beyond retrieval, we further provide preliminary evidence that CAST can serve as a useful reranking signal for selecting more coherent candidates from black-box video generation models (e.g., Veo).

In summary, our contributions are threefold:

- We formalize Consistent Video Retrieval (CVR) and introduce a diagnostic benchmark that isolates state and identity consistency through hard negatives.

- We propose CAST, a lightweight trainable adapter that predicts latent state transitions $\Delta$ to support causally plausible retrieval under procedural context.

- Beyond retrieval, we provide preliminary evidence that CAST can serve as a useful reranking signal for black-box video generation candidates (e.g., from Veo).

**Conflict of Interest Disclosure.** Some authors are employed by Google, which develops Veo, a video generation model used in the black-box generation reranking study in Section 3.4.

## 2. Method

We first introduce our diagnostic benchmark protocol (Figure 2; §2.2) for evaluating state and identity consistency in retrieval. We then present CAST (Figure 3; §2.3), a

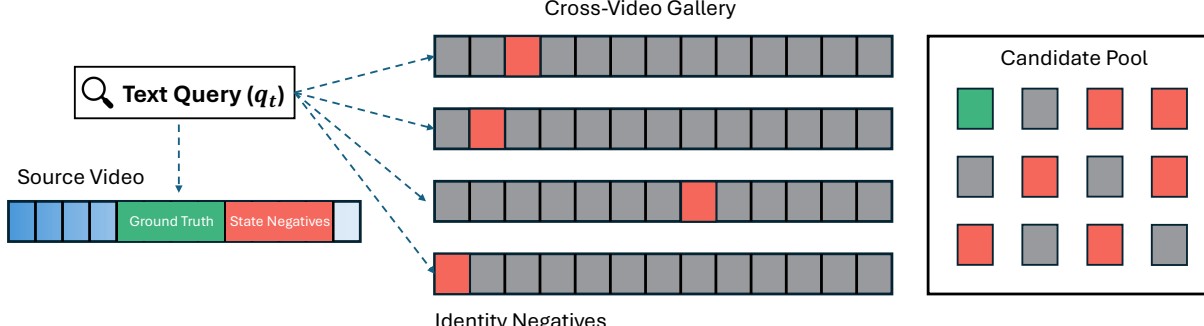

*Figure 2.* **Illustration of our CVR benchmark protocol.** In contrast to standard global retrieval, our benchmark introduces **State Negatives** (temporally misaligned clips from the same video) and **Identity Negatives** (appearance-misaligned clips from different videos) to diagnose consistency failures beyond semantic matching.

lightweight state-transition adapter that addresses these failure modes via residual transition modeling.

### 2.1. Problem Formulation

**Standard Text-to-Video Retrieval.** Given a video gallery $\mathcal{V} = \{v_i\}_{i=1}^N$ and a text query $q$, standard retrieval seeks the clip $v^* \in \mathcal{V}$ with the highest semantic similarity to $q$. Formally, this amounts to maximizing similarity in the joint embedding space:

$$v^* = \underset{v \in \mathcal{V}}{\operatorname{argmax}} \ \operatorname{sim}(f_t(q), f_v(v)), \qquad (1)$$

where $f_t$ and $f_v$ are pre-trained encoders (e.g., CLIP), and $\operatorname{sim}(\cdot)$ denotes a similarity metric such as cosine similarity. Equivalently, this formulation models $P(v \mid q)$. Crucially, each query is treated independently. In procedural narratives, however, this independence assumption is fundamentally ill-suited: it ignores the preceding visual context and therefore may retrieve clips that are semantically relevant to the query yet inconsistent with the evolving event state.

**Consistent Video Retrieval.** To overcome the structural limitation of standard retrieval, we reformulate the task as a sequential, context-aware inference problem in which the target clip $v_t$ is conditioned on both the current instruction $q_t$ and the visual history. Let $\mathcal{H}_t = \{v_{t-L}, \ldots, v_{t-1}\}$ denote the recent narrative history preceding step $t$, where $L$ is the context window length. Our objective is to maximize the conditional probability:

$$v_t^* = \underset{v \in \mathcal{V}}{\operatorname{argmax}} \ P(v \mid \mathcal{H}_t, q_t). \qquad (2)$$

Eq. 2 imposes two constraints: $v_t^*$ must remain locally aligned with the instruction $q_t$ while also preserving state and identity consistency with $\mathcal{H}_t$. In practice, we instantiate Eq. 2 by predicting a target embedding $\hat{v}_t$ and comparing it against candidates in the shared embedding space. This shift from context-agnostic matching in Eq. 1 to context-aware inference in Eq. 2 motivates CAST.

### 2.2. Benchmark Construction

**Limitations of Existing Benchmarks.** Standard video retrieval benchmarks, such as MSR-VTT (Xu et al., 2016), MSVD (Chen & Dolan, 2011), and DiDeMo (Anne Hendricks et al., 2017), typically adopt a global retrieval protocol that ranks each query against the full video gallery. Because most gallery clips differ substantially in semantic content, models can perform well by relying on coarse-grained visual cues, such as objects or scenes, without modeling temporal consistency. As a result, these benchmarks are insufficiently diagnostic: they do not explicitly penalize *state* or *identity* inconsistencies.

**Procedural Data and Context.** To address this gap, we construct the CVR benchmark using three procedural datasets: YouCook2 (Zhou et al., 2018), COIN (Tang et al., 2019), and CrossTask (Zhukov et al., 2019). These datasets exhibit strong causal dependencies ($step_{t-1} \rightarrow step_t$), making them well-suited for evaluating temporal consistency. CrossTask further broadens the evaluation scope beyond cooking. We construct samples using a sliding-window strategy. Annotated step segments serve as the basic clip units, and the corresponding step text is treated as $q_t$. For each query step $q_t$, the context $\mathcal{H}_t$ consists of a variable-length sequence of preceding clips, capped by a maximum window length $L$. This design allows us to evaluate robustness under diverse context lengths.

**Hard Negative Construction.** We formulate evaluation as a multiple-choice ranking task. For each query, we construct a candidate pool $\mathcal{C}$ containing the ground-truth clip and a set of mined negatives. To explicitly diagnose consistency failures, we partition negatives into three types:

- **State Negatives (Temporal Inconsistency):** Sampled from the *same* video but from different non-target step segments. These clips preserve the environment and actor identity but correspond to an invalid procedural state, including both past and future steps.
- **Identity Negatives (Appearance Inconsistency):**

Sampled from *different* videos (excluding the source video) using dataset-specific semantic or structural matching rules. For YouCook2, we mine semantically similar captions using Sentence-BERT. For COIN and CrossTask, we use task and step annotations to construct semantically matched cross-video distractors. These negatives remain semantically relevant but violate identity consistency.

- **Easy Negatives:** Randomly sampled clips with low semantic similarity, used to maintain a fixed candidate-pool size.

**Formal Protocol.** Formally, the candidate set is defined as $\mathcal{C} = \{v_{gt}\} \cup \mathcal{N}_{\text{state}} \cup \mathcal{N}_{\text{identity}} \cup \mathcal{N}_{\text{easy}}$, and performance is evaluated by ranking candidates within this pool. Unless otherwise specified, we adopt a fixed 1-vs-9 protocol, in which each query is evaluated against one ground-truth clip and nine negatives. We first sample up to 3 state negatives and up to 3 identity negatives. If one hard-negative pool is insufficient, the remaining slots are backfilled from the other hard-negative pool; any leftover slots are then filled with easy negatives, keeping the total candidate pool size fixed at 10. This composition is used only for evaluation; training-time candidate construction is described separately in Appendix B. By construction, semantic matching alone is insufficient: to identify the correct target, the model must satisfy both state and identity consistency. We report Recall@K within the candidate pool, and optionally provide results stratified by negative type. In this way, the protocol isolates consistency errors while controlling for semantic difficulty. Additional dataset-specific construction details are provided in §3 and the appendix.

## 2.3. CAST: Context-Aware State Transition

Building on the formulation in §2.1 and the benchmark design in §2.2, we propose **CAST**, a lightweight query-side adapter atop a frozen backbone that models procedural steps as state-conditioned transitions in the embedding space. In contrast to standard dual-encoders that statically match text to video, CAST explicitly predicts the next visual state through a residual update.

**Overview.** CAST is implemented as a lightweight adapter on top of a frozen pre-trained video encoder (e.g., CLIP), while remaining plug-and-play at inference time. Given the anchor state $v_{t-1}$, the instruction $q_t$, and the visual history $\mathcal{H}_t$, the goal is to predict the target representation $\hat{v}_t$. We formulate this prediction as a residual transition:

$$\hat{v}_t = \phi(v_{t-1} + \Delta(v_{t-1}, q_t, \mathcal{H}_t)), \qquad (3)$$

where $\Delta \in \mathbb{R}^d$ denotes the predicted transition vector and $\phi(\cdot)$ denotes L2 normalization.

**Why Simple Residual Modeling Works.** The central intuition behind CAST is that procedural actions (e.g., "slice the

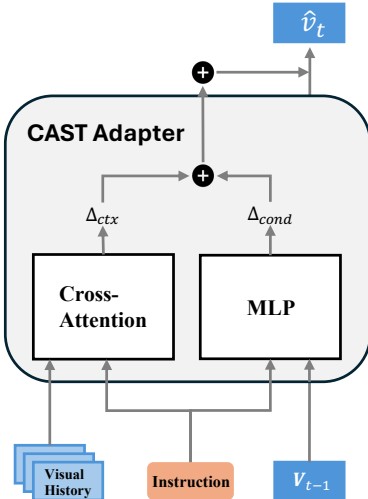

*Figure 3.* **Overview of the CAST adapter.** CAST operates as a lightweight adapter that aggregates visual history $\mathcal{H}_t$, anchor state $v_{t-1}$, and instruction $q_t$. Through a dual-path transition predictor, it estimates a residual update $\Delta$ that encourages causally consistent state evolution while retaining identity cues through the residual connection.

tomato") primarily alter state-related visual factors, while much of the scene, including identity and background, remains persistent. Modeling the transition $\Delta$ additively therefore introduces a useful inductive bias: the residual branch can focus on procedural change, while the anchor state $v_{t-1}$ carries persistent identity cues through the residual connection. The type-aware contrastive objective further encourages the predicted update to emphasize state-relevant changes rather than static appearance cues, echoing the intuition of content-motion decomposition (Villegas et al., 2017) without additional complexity.

**Dual-Path Transition Prediction.** To capture both local state change and broader narrative context, we decompose the transition vector into two complementary components, $\Delta = \Delta_{cond} + \Delta_{ctx}$.

*1) Path 1: Instruction-Conditioned State Transition.* In contrast to standard approaches that model actions purely from text, CAST grounds the instruction in the current visual state. Concretely, we concatenate the text embedding $f_t(q_t)$ with the anchor visual state $v_{t-1}$ and process the result through an MLP:

$$\Delta_{cond} = \text{MLP}_{cond}([f_t(q_t); v_{t-1}]). \qquad (4)$$

This branch predicts how the specific scene encoded by $v_{t-1}$ should evolve under instruction $q_t$, thereby tightly coupling action semantics with the current observation.

*2) Path 2: Temporal Context Attention.* To resolve ambiguities that depend on longer-term history (e.g., distinguishing "stir" in early versus late cooking stages), we employ a multi-head cross-attention mechanism (Vaswani et al., 2017). The

instruction embedding $f_t(q_t)$ serves as the query, while the visual history sequence $\mathcal{H}_t$ provides the keys and values. We first compute a context feature $z_{\text{ctx}}$ using multi-head attention, and then project it with an MLP:

$$z_{\text{ctx}} = \text{MultiHead}\left(f_t(q_t), \mathcal{H}_t, \mathcal{H}_t\right), \tag{5}$$

$$\Delta_{\text{ctx}} = \text{MLP}_{\text{ctx}}(z_{\text{ctx}}). \tag{6}$$

This branch aggregates relevant historical cues into a context-aware adjustment vector $\Delta_{\text{ctx}}$.

**State as an Emergent Property.** Importantly, CAST does not assume a discrete or pre-defined state space. Instead, it treats visual state as an emergent property of the embedding geometry shaped by procedural supervision. By modeling $\Delta$ as an instruction-conditioned vector shift, rather than as a scalar reweighting of existing features, CAST predicts a target embedding that extrapolates the current state toward a plausible next step. This notion is operationalized through measurable gains in retrieval consistency under the controlled hard-negative settings of our CVR protocol, without requiring task-specific state annotations.

Detailed architectural specifications, including layer dimensions, attention configuration, normalization, and padding or masking strategy, are provided in Appendix B.

### 2.4. Training and Inference

**Type-Aware Contrastive Objective.** We freeze the backbone and optimize only the CAST adapter. For each query step, CAST predicts the next-state embedding $\hat{v}_t$ from the instruction and visual history. Let $v_t^+$ denote the ground-truth continuation embedding, $\mathcal{N}_s(t)$ the mined state negatives, and $\mathcal{N}_i(t)$ the mined identity negatives. We use cosine similarity $\text{sim}(\cdot, \cdot)$ with temperature $\tau$.

We begin with a standard in-batch InfoNCE objective for global discrimination:

$$\mathcal{L}_{\text{batch}} = -\frac{1}{B} \sum_{t=1}^{B} \log \frac{a_t}{\sum_{j=1}^{B} a_{t,j}}, \tag{7}$$

where

$$a_t = \exp\left(\text{sim}(\hat{v}_t, v_t^+)/\tau\right), \quad a_{t,j} = \exp\left(\text{sim}(\hat{v}_t, v_j^+)/\tau\right). \tag{8}$$

To explicitly enforce fine-grained consistency beyond global batch discrimination, we further introduce two local contrastive objectives:

$$\mathcal{L}_{\text{state}} = -\frac{1}{B} \sum_{t=1}^{B} \log \frac{a_t}{a_t + \sum_{v \in \mathcal{N}_s(t)} b_t(v)}, \tag{9}$$

$$\mathcal{L}_{\text{ident}} = -\frac{1}{B} \sum_{t=1}^{B} \log \frac{a_t}{a_t + \sum_{v \in \mathcal{N}_i(t)} b_t(v)}, \tag{10}$$

where

$$b_t(v) = \exp(\text{sim}(\hat{v}_t, v)/\tau). \tag{11}$$

Rather than aggregating state and identity negatives into a single denominator, we optimize $\mathcal{L}_{\text{state}}$ and $\mathcal{L}_{\text{ident}}$ separately. This prevents highly overlapping identity negatives from dominating the gradients, thereby preserving a stronger signal for fine-grained state discrimination.

The final objective is:

$$\mathcal{L} = \mathcal{L}_{\text{batch}} + \lambda_s \mathcal{L}_{\text{state}} + \lambda_i \mathcal{L}_{\text{ident}}. \tag{12}$$

Here, $\lambda_s > \lambda_i$ reflects our design prior that fine-grained state discrimination constitutes the primary challenge in CVR. At the same time, the residual prediction design anchors the transition around $v_{t-1}$, which already provides a useful bias toward retaining identity-consistent context.

**Plug-and-Play Inference.** At inference time, CAST operates as a scalable query-side plug-and-play module. The video gallery is pre-indexed once using the frozen backbone. Given a query step, CAST computes $\hat{v}_t$ on-the-fly from the visual history and instruction, and scores candidate continuations in the same frozen embedding space.

For a query instruction $q$ and a candidate clip $c$, we define three scores:

$$A(q, c) = \text{sim}(t_q, v_c), \tag{13}$$

$$B(q, c) = \text{sim}(v_{t-1}, v_c), \tag{14}$$

$$C(q, c) = \text{sim}(\hat{v}_t, v_c), \tag{15}$$

where $t_q$ is the query text embedding, $v_{t-1}$ is the last context-clip embedding, $\hat{v}_t$ is the CAST-predicted next-state embedding, and $v_c$ is the candidate clip embedding. Here, $A$ captures semantic alignment, $B$ visual continuity with the anchor state, and $C$ compatibility with the predicted future state.

We combine these signals using the *Full Ensemble*:

$$S(q, c) = A(q, c) + w_v B(q, c) + w_p C(q, c). \tag{16}$$

The ensemble coefficients are selected on a held-out validation split for each dataset and backbone setting, and then kept fixed during evaluation. The coefficient selection protocol and search ranges are provided in Appendix B. Because CAST operates exclusively on the query side and does not require re-indexing the video gallery, it preserves the scalability of standard retrieval pipelines while enabling retrieval under temporal context.

## 3. Experiments

### 3.1. Experimental Setup

**Datasets and Evaluation Protocol.** We evaluate CAST on three procedural video datasets: YouCook2 (Zhou et al.,

*Table 1.* **CVR benchmark statistics.** We report the size of the final CVR evaluation set for each dataset. All settings follow the fixed 1-vs-9 ranking protocol described in Sec. 2.2. Dataset-specific negative mining rules are summarized in Appendix Table 9.

| Dataset | Split Protocol | #Videos | #Step-Clips | #Queries |
|---|---|---|---|---|
| YouCook2 (Zhou et al., 2018) | official train; CVR eval on val | 414 | 3,179 | 2,765 |
| COIN (Tang et al., 2019) | official train; CVR eval on test | 2,134 | 6,241 | 4,107 |
| CrossTask (Zhukov et al., 2019) | video-disjoint 80/20 split | 509 | 2,731 | 2,222 |

2018), COIN (Tang et al., 2019), and CrossTask (Zhukov et al., 2019), with benchmark statistics summarized in Table 1. For YouCook2 and COIN, we train on the official training splits and evaluate on the official validation/test splits, respectively. For CrossTask, we use a video-disjoint 80/20 split. All final metrics are computed on held-out evaluation videos, and any validation subset used for ensemble coefficient selection is drawn only from the training portion.

For CVR evaluation, we use a fixed 1-vs-9 ranking protocol: each query is evaluated against one ground-truth continuation and nine negatives, including state negatives, identity negatives, and easy negatives. Dataset-specific mining rules, training-time candidate construction, and coefficient-selection details are provided in Appendix B.

**Baselines.** We compare CAST against four baselines: (i) **CLIP Baseline**, a zero-shot context-free matching formulation using the frozen backbone; (ii) **Late Fusion (Heuristic)**, a fixed weighted sum of semantic and visual scores $(S = sim(q,v) + \alpha \cdot sim(v_{t-1}, v))$; (iii) **Late Fusion (Learned)**, a lightweight MLP trained to learn dataset-specific weighting over semantic and visual scores, serving as a strong aggregation baseline; and (iv) **Early Fusion**, a trainable MLP that concatenates text and context features before prediction.

**Implementation Details.** For the main experiments, we use a frozen CLIP ViT-B/32 backbone and train only the CAST adapter with AdamW. We use batch size 512, context window $L = 5$, and loss weights $\lambda_s = 5.0$, $\lambda_i = 1.0$. Unless otherwise specified, CAST uses the Full Ensemble score in Eq. (16), with coefficients selected on a held-out validation subset and fixed during evaluation. Full optimization details, hard-negative construction, backbone-specific feature extraction, and inference settings are provided in Appendix B and Appendix A.

**Metrics.** We report **Accuracy (Acc.)** (Recall@1) and **Mean Rank (MnR)**. To diagnose failure modes more precisely, we define **State Acc.** (the ground-truth clip ranked above all State Negatives) and **Ident. Acc.** (the ground-truth clip ranked above all Identity Negatives). Diagnostic accuracies are computed over queries containing at least one negative of the corresponding type. For a global assessment of state discrimination and identity preservation, the diagnostic scores reported in our main results (Table 2 and

Table 3) are averaged across the three benchmark datasets.

**Control for Semantic Difficulty.** Our 1-vs-9 ranking protocol is designed so that performance gains cannot be attributed solely to vision-language alignment. By constructing each candidate pool from a mixture of state, identity, and easy negatives, we require the model to move beyond keyword matching and leverage the visual context $\mathcal{H}_t$ to resolve temporal ambiguities. This controlled setup makes the improvements reported in Table 2 directly interpretable as evidence of stronger state discrimination and more reliable identity preservation under contextual constraints.

### 3.2. Quantitative Results on CVR Benchmark

We organize our quantitative analysis around two central questions: (1) Does explicit state-transition modeling outperform standard context aggregation? (2) Does CAST transfer effectively across diverse frozen foundation-model backbones?

**1. Effectiveness of the CAST Mechanism.** In Table 2, we fix the backbone to CLIP (ViT-B/32) in order to isolate the contribution of the adapter architecture itself.

**Limitations of Scalar Aggregation.** To test whether simple score re-weighting is sufficient, we include a strong Late Fusion (Learned) baseline. On COIN, where consecutive procedural steps often exhibit strong visual similarity, the learned fusion can exploit visual inertia (Geirhos et al., 2020), achieving its best performance (44.66%) and slightly surpassing CAST (40.47%). However, this shortcut does not generalize to datasets with more substantial state transitions, such as CrossTask and YouCook2, where procedural steps involve more pronounced visual changes (e.g., "add ginger paste" → "fry"). In these settings, the learned baseline degrades sharply, dropping to 25.52% on CrossTask, which is 21.9 points below CAST in accuracy.

**Vector Transition vs. Scalar Weighting.** The cross-dataset performance gap highlights a fundamental difference between CAST and aggregation-based baselines. Whereas scalar late-fusion models can only reweight existing similarity signals, CAST predicts a structured, instance-conditioned residual $\Delta$ in the latent space. This formulation is more effective at resolving causal ambiguities, as evidenced by CAST's consistently stronger State Accuracy under the diagnostic protocol.

**2. Universality Across Backbones.** While Table 2 isolates

*Table 2.* **CVR Benchmark Results (CLIP-B/32).** CAST provides the most favorable overall trade-off across datasets and diagnostic metrics, with especially clear gains on state-sensitive retrieval. Unless otherwise specified, CAST uses the Full Ensemble score defined in Eq. (16). Diagnostic scores (State Acc. and Ident. Acc.) are averaged across the three benchmark datasets.

| Method | Context Modeling | YouCook2 Acc.↑ | YouCook2 MnR↓ | COIN Acc.↑ | COIN MnR↓ | CrossTask Acc.↑ | CrossTask MnR↓ | Diagnostic State↑ | Diagnostic Ident.↑ |
|---|---|---|---|---|---|---|---|---|---|
| CLIP Baseline | Context-Free | 25.03 | 3.60 | 14.10 | 3.91 | 16.83 | 4.15 | 45.52 | 28.90 |
| Late Fusion (Heuristic) | Fixed Weighting | 31.10 | 2.56 | 17.85 | 3.28 | 22.05 | 2.86 | 28.69 | 68.29 |
| Late Fusion (Learned) | Learned Weighting | 36.60 | 2.53 | **44.66** | **2.11** | 25.52 | 2.86 | 40.06 | 76.06 |
| Early Fusion | Feature Concat. | 35.99 | 2.28 | 15.12 | 2.60 | 35.29 | 2.36 | 31.14 | **83.59** |
| **CAST (Ours)** | **State Transition** | **44.77** | **2.15** | 40.47 | 2.16 | **47.39** | **2.14** | **53.81** | 74.67 |

*Table 3.* **Universality Across Backbones.** CAST consistently improves over the corresponding zero-shot baseline across diverse frozen video and multimodal embedding models, with both methods operating in the same frozen native vision-language embedding space for each backbone. Diagnostic scores are averaged across the three benchmark datasets.

| Backbone | Setting | YouCook2 Acc. | YouCook2 MnR | COIN Acc. | COIN MnR | CrossTask Acc. | CrossTask MnR | Diagnostic State | Diagnostic Ident. |
|---|---|---|---|---|---|---|---|---|---|
| *Category I: Video Foundation Models* | | | | | | | | | |
| InternVideo2-1B (Wang et al., 2024) | Zero-Shot | 36.75 | 2.59 | 17.99 | 3.36 | 20.61 | 3.31 | 65.70 | 30.85 |
| | + CAST | **71.68** | **1.48** | **51.03** | **1.90** | **64.36** | **1.71** | **75.43** | **77.77** |
| VideoPrism-B (Zhao et al., 2024) | Zero-Shot | 47.45 | 2.13 | 17.60 | 3.32 | 20.25 | 3.24 | 68.38 | 33.68 |
| | + CAST | **75.59** | **1.38** | **51.64** | **1.90** | **62.11** | **1.74** | **76.92** | **77.66** |
| *Category II: Multimodal Embedding Models* | | | | | | | | | |
| GME-Qwen2-VL-2B (Zhang et al., 2024) | Zero-Shot | 29.62 | 3.10 | 17.17 | 3.44 | 19.40 | 3.61 | 56.31 | 29.73 |
| | + CAST | **54.39** | **1.95** | **45.68** | **2.05** | **52.43** | **2.04** | **67.20** | **72.28** |
| Qwen3-VL-Embedding-2B (Li et al., 2026) | Zero-Shot | 33.45 | 2.89 | 17.73 | 3.50 | 19.44 | 3.56 | 58.44 | 29.79 |
| | + CAST | **56.64** | **1.85** | **44.87** | **2.09** | **48.96** | **2.09** | **66.18** | **69.11** |

CAST under CLIP ViT-B/32, Table 3 evaluates whether the same design transfers across diverse frozen video and multimodal embedding models, including InternVideo2 (Wang et al., 2024), VideoPrism (Zhao et al., 2024), GME-Qwen2-VL-2B (Zhang et al., 2024), and Qwen3-VL-Embedding (Li et al., 2026). For each backbone, both the zero-shot baseline and CAST operate in the same native frozen embedding space, while only the query-side CAST adapter is trained.

The gains are consistent across backbone families. On InternVideo2, CAST improves accuracy from 36.75% to 71.68% on YouCook2 and from 20.61% to 64.36% on CrossTask. Similar gains hold for VideoPrism and multimodal embedding models, e.g., GME-Qwen2-VL-2B improves by +24.6% on YouCook2, and Qwen3-VL-Embedding improves from 33.45% to 56.64%. These results suggest that CAST transfers across stronger frozen representation spaces. Backbone checkpoints, preprocessing, and feature extraction details are provided in Appendix A.

**Identity Consistency and Embedding Quality.** Across all backbones, CAST consistently improves identity preservation under hard identity negatives, raising Ident. accuracy from roughly 30% to 69–78% (Table 3). Notably, the gains scale with the quality of the underlying embedding space, suggesting that the same CAST design transfers effectively

across different frozen spaces while more fully exploiting stronger visual representations.

**Qualitative Retrieval Examples.** To complement the metrics, Figure 4 presents two representative CVR retrieval cases. In both examples, the context-agnostic baseline retrieves a semantically related clip but fails to preserve either the current procedural state or the visual identity. By contrast, CAST retrieves the correct continuation by explicitly modeling the underlying state transition.

### 3.3. Ablation Studies

To isolate the sources of CAST's gains, we conduct ablation studies over both the architecture design (Table 4) and the inference formulation (Table 5).

**1. Architecture and Objective Analysis.** Table 4 presents a factorial analysis of our design choices on the YouCook2 validation benchmark. First, at the objective level, comparing the first two rows shows that *Residual Modeling* ($\Delta$) is critical. Replacing direct target prediction with an additive residual update improves the Early Fusion model from 35.99% to 38.95%, while raising State Accuracy from 38.92% to 43.51%. This result supports the central inductive bias of CAST: anchoring prediction around $v_{t-1}$ makes pro-

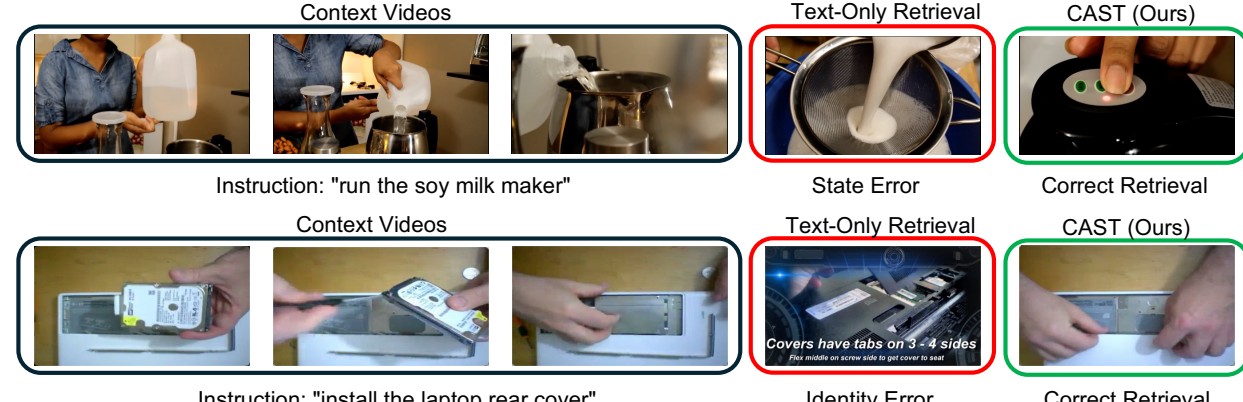

Figure 4. **Qualitative retrieval examples on the CVR benchmark.** Given the same procedural context and instruction, context-agnostic retrieval often returns semantically relevant but temporally inconsistent clips, producing either a *State Error* or an *Identity Error*. By contrast, CAST retrieves the correct continuation in both cases by modeling state transitions conditioned on visual history.

Table 4. **Ablation of target formulation and architecture on the YouCook2 validation benchmark.** Residual modeling consistently outperforms direct target prediction, and the dual-path CAST design further improves over simple early fusion.

| Fusion | Target | Acc. | State | Ident. |
|--------|--------|------|-------|--------|
| Early (Concat) | Direct ($\hat{v}_t$) | 35.99 | 38.92 | **83.58** |
| Early (Concat) | Residual ($\Delta$) | 38.95 | 43.51 | 81.99 |
| **CAST (Ours)** | **Residual ($\Delta$)** | **44.77** | **51.03** | 78.48 |

Table 5. **Inference signal decomposition on the YouCook2 validation benchmark.** We analyze the three inference components defined in Sec. 2.4 and their combinations. While the semantic ensemble achieves the highest exact-match accuracy, the full ensemble provides a more balanced trade-off between state discrimination and identity preservation.

| Inference Strategy | Acc. | State | Ident. |
|--------------------|------|-------|--------|
| A. Text Matching ($q$) | 25.03 | 50.45 | 25.32 |
| B. Vis. Continuity ($v_{t-1}$) | 25.90 | 27.70 | **81.95** |
| C. CAST Prediction ($\hat{v}_t$) | 42.60 | 50.81 | 75.99 |
| *Semantic Ens. (A+C)* | **45.46** | **56.71** | 70.38 |
| **Full Ens. (A+B+C)** | 44.77 | 51.03 | 78.48 |

cedural continuation easier to model, because the adapter only needs to predict a targeted state update rather than reconstruct the full next-step embedding from scratch.

Second, at the architectural level, comparing *Early Fusion + Residual* with CAST (Dual-Path + Residual) reveals a gain of +4.05% Acc. This indicates that simple feature concatenation is insufficient for modeling noisy procedural context. By contrast, CAST's dual-path design and cross-attention mechanism provide a more effective way to isolate temporally relevant cues from visual history.

**2. Inference Components.** Table 5 decomposes the three inference components defined in Sec. 2.4: the semantic

matching score $A$, the visual continuity score $B$, and the CAST prediction score $C$, together with their *Full Ensemble* combination in Eq. (16).

Several trends are apparent. Semantic matching alone improves recall but severely degrades identity preservation. Among the individual components, CAST prediction ($C$) already provides the most balanced performance, indicating that the predicted future-state embedding captures a meaningful procedural continuation signal. While the Semantic Ensemble ($A+C$) attains the highest overall accuracy, adding visual continuity ($B$) improves identity preservation from 70.38 to 78.48, with only a modest drop in exact-match accuracy. We therefore adopt the Full Ensemble as the default inference strategy, since it provides the most balanced overall trade-off between state discrimination and identity preservation. A qualitative breakdown of retrieval outcomes is provided in Appendix B.2.

**3. Context History Length.** We further vary the context window length $L$ and find that performance improves sharply from the context-free setting ($L = 0$) to using the immediate predecessor ($L = 1$), then largely saturates as longer history is added. This suggests that $v_{t-1}$ serves as the primary causal anchor. Full results are provided in Appendix B.1.

### 3.4. Application: Guiding Video Generation

Beyond retrieval, CAST can also rerank black-box video generation candidates using its predicted transition compatibility. Given a single visual context image, extracted as the last frame of the most recent context clip, together with a next-step instruction, we use Veo (Google DeepMind, 2024) to generate $K = 4$ candidate continuation videos for each prompt. We then compare two ranking strategies over the same candidate pool: **Standard Text Match**, which uses a text-only retrieval score, and **CAST Reranking**, which

*Table 6.* **Guiding Video Generation.** Blind evaluation on 300 YouCook2 validation prompts. We compare the top-ranked videos selected by standard text matching and CAST reranking. Results are majority-vote preference rates on the 158 non-overlap cases.

| Method | Overall Pref. | Physical Plaus. | Temporal Logic |
|---|---|---|---|
| Text Match | 38.6% | 39.9% | 38.6% |
| **CAST** | **55.1%** | **50.6%** | **52.5%** |
| Human Tie | 6.3% | 9.5% | 8.9% |

applies the same final reranking strategy used in retrieval.

We conduct a blind human study on 300 prompts sampled from the YouCook2 validation benchmark. No training-set queries are used in the generation or human-evaluation protocol. For each prompt, standard text matching and CAST reranking each select one top-ranked video from the same generated candidates. The two methods select different candidates in 158 out of 300 prompts, yielding the non-overlap subset for pairwise comparison. Equivalently, CAST changes the final selected candidate on 158/300 prompts, defining the subset on which pairwise comparison is informative. We compare the two selected outputs in randomized order using three annotators and aggregate the final label by majority vote. As shown in Table 6, CAST-selected outputs are preferred over standard text matching across overall preference, physical plausibility, and temporal logic. Additional setup, annotation protocol, and qualitative examples are provided in Appendix C.

## 4. Related Work

**Video-Text Retrieval.** Video-text retrieval is dominated by dual-encoder architectures (Radford et al., 2021) and generative-contrastive learners such as VideoCoCa (Yan et al., 2022), which map clips and queries into a shared embedding space (Bain et al., 2021; Luo et al., 2021; Wang et al., 2022; Zhao et al., 2024). Early temporal models such as VideoBERT (Sun et al., 2019) used discretized visual tokens, while later models including HERO (Li et al., 2020), UniVL (Luo et al., 2020), and VIOLET (Fu et al., 2021) introduce richer temporal or cross-modal modeling. However, these methods still primarily optimize global video-text alignment. In contrast, CAST targets retrieval under procedural context, using an extrapolated visual state as the retrieval anchor to enforce consistency beyond standard feature matching.

**Procedural Video Understanding.** Goal-oriented procedures require modeling causal dependencies across steps (Zhou et al., 2018; Tang et al., 2019; Zhukov et al., 2019). Prior work has studied dense captioning (Wang et al., 2018), procedural planning (Chang et al., 2020), and visual state modeling, including state discovery and state-change detection (Isola et al., 2015; Souček et al., 2022). These approaches often represent state changes with discrete categories, localized cues, or planning objectives, whereas CAST targets *Consistent Video Retrieval* (CVR): retrieving a semantically relevant clip that is also consistent with the preceding visual state and identity.

**Connection to World Models.** Predictive world models learn representations by forecasting future states in latent space (Bardes et al., 2024; Assran et al., 2025). CAST shares this future-state perspective, but focuses on *instruction-conditioned retrieval*: query text specifies the intended transition, and predicted visual state is used as a retrieval anchor. Following representation learning principles (Bengio et al., 2013), we treat procedural state as an *emergent geometric property* of the embedding space, helping bias CAST toward task-relevant progression while reducing reliance on static visual shortcuts (Geirhos et al., 2020).

## 5. Limitations

CAST remains subject to several limitations. First, the current context window is fixed ($L = 5$). While sufficient for short procedural sequences, it remains limited in capturing longer-range dependencies, such as objects or state changes introduced several steps earlier. Second, as a lightweight adapter, CAST is inherently constrained by the representational quality of the frozen backbone. When the base encoder fails to resolve subtle state differences, such as fine-grained texture variations or object-configuration changes, CAST is correspondingly limited. Third, the residual transition is learned without explicit geometric constraints on $\Delta$. Although this formulation is effective empirically, it does not explicitly constrain the latent space to separate temporal progression from persistent identity cues. Promising directions for future work include hierarchical memory for longer-horizon reasoning and more structured regularization for state-transition modeling.

## 6. Conclusion

We introduced **Consistent Video Retrieval (CVR)** to address a structural limitation of standard retrieval systems in maintaining temporal and identity coherence. We proposed CAST, a lightweight adapter that models procedural steps as state-conditioned residual transitions. Extensive experiments on YouCook2, COIN, and CrossTask show that CAST improves retrieval under procedural context, yielding clear gains on YouCook2 and CrossTask, remaining competitive on COIN, and consistently improving over the corresponding zero-shot baselines across diverse frozen embedding backbones. Beyond retrieval, we show that CAST can rerank candidate videos generated by black-box video generation models toward more coherent continuations. We hope this work encourages further research on context-aware inference and causally consistent video understanding.

## Impact Statement

This work introduces a benchmark and method for improving temporal and identity consistency in video retrieval and generation reranking. Potential positive impacts include supporting more coherent video organization, editing, and generation workflows. Potential risks include misuse in automated media selection or generation pipelines, where improved coherence may make synthetic video content more persuasive. We encourage responsible use with clear provenance, human oversight, and appropriate disclosure when applied to generated media.

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

# A. Appendix: Backbone Details for Universality Experiments

This appendix documents the frozen backbones used in the *Universality Across Backbones* experiments, together with the exact feature extraction and preprocessing protocols, to facilitate reproducibility.

**General Protocol.** For all backbone variants, we use publicly available pretrained checkpoints and follow the authors' official inference or feature extraction pipelines. All backbones remain frozen throughout, and no task-specific fine-tuning is applied. For the universality results in Table 3, each backbone operates in its own native frozen text-video embedding space. Accordingly, both the zero-shot baseline and CAST use the backbone's native text encoder for queries and native video encoder for clips. CAST retains the same architecture, training objective, and evaluation protocol as in the CLIP-based experiments, while operating in the corresponding backbone-specific frozen text-video embedding space.

Training schedules depend on both dataset and backbone. For the main CLIP-based experiments, we train for 30 / 20 / 50 epochs on YouCook2 / COIN / CrossTask, respectively. For the alternative frozen backbones (InternVideo2, VideoPrism, GME-Qwen2-VL-2B, and Qwen3-VL-Embedding-2B), we train for 30 / 30 / 50 epochs on YouCook2 / COIN / CrossTask, respectively.

Frame sampling is likewise backbone-dependent. CLIP, GME-Qwen2-VL-2B, and Qwen3-VL-Embedding-2B use 3 frames per clip on all datasets. InternVideo2 uses 8 frames per clip on all datasets. VideoPrism uses 8 frames per clip on YouCook2 and 4 frames per clip on COIN and CrossTask. Frame-level features are aggregated into clip-level embeddings via mean pooling. All extracted clip embeddings are L2-normalized prior to retrieval and CAST training.

**Text Side and Score Definitions in Universality Experiments.** For the universality results in Table 3, both the zero-shot baseline and CAST are evaluated within each backbone's native frozen text-video embedding space. For video foundation models such as InternVideo2 and VideoPrism, query text embeddings are obtained from the backbone's own frozen text encoder, and clip embeddings are obtained from the corresponding frozen video encoder. For multimodal embedding backbones (e.g., GME-Qwen2-VL-2B and Qwen3-VL-Embedding-2B), both text and video embeddings are produced directly by the same pretrained multimodal embedding model.

Accordingly, the three inference scores are all defined within the corresponding backbone space. The semantic score $A$ denotes the native text-video similarity between the query text embedding and a candidate clip embedding. The visual continuity score $B$ denotes the similarity between the last context clip embedding and the candidate clip embedding. The prediction score $C$ denotes the similarity between the CAST-predicted next-state embedding and the candidate clip embedding. Thus, Table 3 compares zero-shot retrieval and CAST reranking within the same frozen backbone-specific embedding space, rather than under a shared text encoder.

**InternVideo2 (PyTorch).** We use the official InternVideo2 Stage-2 inference code and pretrained checkpoint. Each clip is represented by 8 uniformly sampled frames. Frames are resized to $224 \times 224$, normalized using ImageNet mean/std, and packed into a tensor of shape $[B, T, C, H, W]$. We extract clip embeddings using the model's video feature head (e.g., `get_vid_feat`), keep the backbone frozen, and store one embedding per clip. For efficiency, we perform extraction with multi-GPU data parallelism, shard the video list by rank, save per-rank feature dictionaries, and merge them offline.

**VideoPrism (JAX/Flax).** We use the public VideoPrism LVT checkpoint together with the official Flax model and weight loader. Each clip is represented by 8 uniformly sampled frames on YouCook2 and 4 uniformly sampled frames on COIN and CrossTask, with all frames resized to $288 \times 288$. Pixels are scaled to $[0, 1]$ and provided in NHWC format, as required by the JAX implementation. Since the public VideoPrism forward interface expects text inputs, we pass dummy text IDs and paddings during video feature extraction and retain only the video embeddings. For Table 3, query text embeddings are still obtained from the backbone's native frozen text encoder; the dummy text inputs are used solely to satisfy the public video inference interface. We extract one clip embedding per clip and L2-normalize the resulting vectors for retrieval.

**Multimodal Embedding Models (GME-Qwen2-VL-2B, Qwen3-VL-Embedding).** For multimodal embedding backbones, we use publicly available pretrained checkpoints and operate directly on their embedding outputs without fine-tuning. Each clip is represented by 3 uniformly sampled frames decoded with OpenCV and converted to PIL images. We encode each frame with the embedding model to obtain frame-level embeddings, then mean-pool across frames to form a clip embedding, followed by L2 normalization.

*Table 7.* Backbone feature extraction settings for the universality experiments. All backbones are frozen. For each backbone, zero-shot retrieval and CAST are both evaluated in the backbone's native text-video embedding space, while the table below summarizes the corresponding video-side feature extraction settings.

| Backbone | Impl. | #Frames/clip (YC2 / COIN / CT) | Res. | Post-proc. |
|---|---|---|---|---|
| InternVideo2-1B | PyTorch | 8 / 8 / 8 | 224 | mean pool + L2 norm |
| VideoPrism-B | JAX/Flax | 8 / 4 / 4 | 288 | mean pool + L2 norm |
| GME-Qwen2-VL-2B | PyTorch | 3 / 3 / 3 | native | mean pool + L2 norm |
| Qwen3-VL-Embedding-2B | PyTorch | 3 / 3 / 3 | native | mean pool + L2 norm |

**Qwen3-VL-Embedding.** We use `Qwen3-VL-Embedding-2B` and extract embeddings via a lightweight wrapper (`Qwen3VLEmbedder`). For efficiency, we run inference in FP16 (`torch_dtype=float16`) with FlashAttention-2 (`attn_implementation=flash_attention_2`). Given a batch of clips, we flatten the 3 frames per clip, compute embeddings for all frames, reshape them back to $[B, 3, D]$, mean-pool over frames to obtain $[B, D]$ clip embeddings, and apply L2 normalization before saving. If decoding fails or a video is corrupted, we pad missing frames with black images to keep the input shape fixed.

## B. Appendix: CAST Training and Inference Details

**Dataset Split and Evaluation Protocol.** We explicitly distinguish three roles for the data in all experiments: (1) the training split used to optimize CAST parameters; (2) a held-out validation subset drawn exclusively from the training portion, used to select the Full Ensemble coefficients; and (3) the final benchmark split used for all reported retrieval metrics.

For YouCook2, training instances are constructed from the official training split, whereas the fixed 1-vs-9 evaluation benchmark is constructed exclusively from the official validation split. For COIN, training instances are constructed from the official training split, whereas the fixed evaluation benchmark is constructed from the official test split. For CrossTask, we first construct a benchmark from the annotated videos and then partition it by unique video ID into disjoint training and evaluation subsets using an 80%/20% split with random seed 42, yielding `crosstask_train.json` and `crosstask_test.json`. All reported CrossTask retrieval metrics are computed only on `crosstask_test.json`, while training uses only `crosstask_train.json`. No source video appears in both training and evaluation.

*Table 8.* **Dataset split protocol used in CAST.**

| Dataset | Weight training | Coefficient tuning | Final benchmark eval |
|---|---|---|---|
| YouCook2 | official train | held-out subset of train | official validation |
| COIN | official train | held-out subset of train | official test |
| CrossTask | `crosstask_train.json` | held-out subset of train | `crosstask_test.json` |

**Benchmark Mining Details.** The evaluation benchmark follows a fixed 1-vs-9 protocol across all datasets, consisting of 1 ground-truth clip and 9 negatives in total. For each query, we first sample up to 3 state negatives and up to 3 identity negatives. If one hard-negative pool is insufficient, we backfill the remaining slots from the other available hard-negative pool. Any remaining slots are then filled with easy negatives to preserve a fixed 9-negative candidate set. We always exclude the ground-truth clip itself from the negative pool, enforce unique candidate IDs within each query, and randomly shuffle the final candidate order to eliminate position bias.

**YouCook2.** We construct the benchmark exclusively from the validation split, retaining only clips with valid video files. For each target step, the context history consists of up to the previous $L$ annotated step segments from the same video. State negatives are sampled from the same video but different step segments, with preference for temporally diverse mismatched steps, e.g., past, future, and additional non-target steps when available, up to 3 in total. When selecting past negatives, we avoid the immediate predecessor whenever possible, since it already appears in the context history. Identity negatives are mined globally across different videos using Sentence-BERT caption similarity. Specifically, we encode all step captions with `all-MiniLM-L6-v2`, compute cosine similarity between captions, and use the ranked cross-video neighbors as the identity-negative pool, from which up to 3 identity negatives are sampled per query. Easy negatives are sampled randomly from different videos.

**COIN.** For COIN, we rely on the annotated procedural structure rather than external text mining. Context history is formed

from preceding steps within the same video. State negatives are sampled from the same video but different steps. Identity negatives are sampled from different videos that share the same recipe/task category and step ID as the target step, yielding semantically matched but identity-inconsistent distractors. Easy negatives are sampled from clips with different recipe/task categories.

**CrossTask.** For CrossTask, context history is again constructed from preceding steps within the same video. State negatives are sampled from the same video but different steps, with preference for past and future mismatched steps when available; as in YouCook2, we avoid using the immediate predecessor as a past negative whenever possible. Identity negatives are sampled from different videos within the same task and same step index as the target step. If insufficient such clips exist, we fall back to clips from different videos within the same task but different step indices. Easy negatives are sampled from different tasks.

Training-time candidate construction differs from the fixed evaluation benchmark above and is described separately below.

**Training Candidate Construction and Hard Negatives.** For training, each instance contains one ground-truth clip together with state and identity hard negatives. We sample up to 3 state negatives and up to 3 identity negatives per instance. If one negative pool is insufficient, we fall back to sampling from the other pool; if both are unavailable, we use zero vectors. This fallback strategy is used only for training-time instance construction. Evaluation instead follows the fixed 1-vs-9 benchmark protocol described in Sec. 2.2 and Sec. 3.1, with up to 3 state negatives and up to 3 identity negatives per query, backfilled across hard-negative pools and then with easy negatives when needed.

*Table 9.* **Dataset-specific benchmark mining rules.**

| Dataset | State negatives | Identity negatives | Easy negatives |
|---|---|---|---|
| YouCook2 | same video, diff step | SBERT top-3 cross-video clips | random diff video |
| COIN | same video, diff step | same recipe + same step, diff video | diff recipe |
| CrossTask | same video, diff step | same task + same step, diff video | diff task |

**CAST Architecture Details.** CAST is implemented as a lightweight residual adapter that operates in the frozen text-video embedding space. Let $d$ denote the embedding dimension of the underlying frozen backbone (e.g., $d = 512$ for CLIP-B/32). Given the query text embedding $q_t \in \mathbb{R}^d$, the last context clip embedding $v_{t-1} \in \mathbb{R}^d$, and a context history $H_t = \{h_1, \ldots, h_L\} \in \mathbb{R}^{L \times d}$, we first apply L2 normalization to all input embeddings.

CAST models the transition as the sum of two complementary paths, $\Delta = \Delta_{\mathrm{cond}} + \Delta_{\mathrm{ctx}}$. The **instruction-conditioned state-transition path** (Path 1 in §2.3) takes the concatenation $[q_t; v_{t-1}] \in \mathbb{R}^{2d}$ as input and applies a two-layer MLP: $\mathrm{Linear}(2d, 2d) \rightarrow \mathrm{LayerNorm}(2d) \rightarrow \mathrm{ReLU} \rightarrow \mathrm{Dropout}(0.1) \rightarrow \mathrm{Linear}(2d, d)$. This yields the transition component $\Delta_{\mathrm{cond}} \in \mathbb{R}^d$.

The context path first projects the query and context features through linear layers, $q'_t = W_q q_t$ and $H'_t = W_h H_t$, where $W_q, W_h \in \mathbb{R}^{d \times d}$. We then apply a single multi-head cross-attention layer with 8 heads, using the projected query as the attention query and the projected context sequence as keys and values. The resulting attended feature is passed through a residual MLP: $\mathrm{LayerNorm}(d) \rightarrow \mathrm{Linear}(d, d) \rightarrow \mathrm{ReLU} \rightarrow \mathrm{Linear}(d, d)$, yielding $\Delta_{\mathrm{ctx}} \in \mathbb{R}^d$. For variable-length context histories, we left-pad the sequence with zeros and apply a key-padding mask in attention.

The final predicted next-state embedding is computed via a direct residual update:

$$\hat{v}_t = \mathrm{Norm}(v_{t-1} + \Delta_{\mathrm{cond}} + \Delta_{\mathrm{ctx}}),$$

where $\mathrm{Norm}(\cdot)$ denotes L2 normalization. Unless otherwise stated, we do not use an additional gating module in the main model.

**Optimization.** We train CAST with AdamW (lr $= 10^{-4}$, weight decay $= 10^{-3}$) using batch size 512. The number of training epochs is both dataset- and backbone-dependent. For the main CLIP-based experiments, we train for 30, 20, and 50 epochs on YouCook2, COIN, and CrossTask, respectively. For the alternative frozen backbones (InternVideo2, VideoPrism, GME-Qwen2-VL-2B, and Qwen3-VL-Embedding-2B), we train for 30, 30, and 50 epochs on YouCook2, COIN, and CrossTask, respectively. We use the Type-Aware contrastive objective defined in Sec. 2.4, with temperature $\tau = 0.07$ and loss weights $\lambda_s = 5.0$ and $\lambda_i = 1.0$.

*Table 10.* **CAST module specification.**

| Item | Specification |
|------|---------------|
| Embedding dimension $d$ | backbone-dependent (512 for CLIP-B/32) |
| Input normalization | L2 normalization |
| Conditioned path | $\mathrm{Linear}(2d, 2d)$ + LN + ReLU + Dropout(0.1) + $\mathrm{Linear}(2d, d)$ |
| Context projections | $\mathrm{Linear}(d, d)$ for query and context |
| Cross-attention | 1 layer, 8 heads |
| Context MLP | LN + $\mathrm{Linear}(d, d)$ + ReLU + $\mathrm{Linear}(d, d)$ |
| Residual update | $\hat{v}_t = \mathrm{Norm}(v_{t-1} + \Delta_{\mathrm{cond}} + \Delta_{\mathrm{ctx}})$ |
| Gating | none |

*Table 11.* **CAST training and inference settings.** Optimizer, batch size, temperature, and the overall inference formulation are shared across experiments, while selected training and inference details vary by dataset and backbone setting.

| Item | Setting |
|------|---------|
| Context length $L$ | 5 (left-pad with zeros; masked attention) |
| Training epochs | CLIP: 30 / 20 / 50 on YouCook2 / COIN / CrossTask; InternVideo2 / VideoPrism / GME-Qwen2-VL-2B / Qwen3-VL-Embedding-2B: 30 / 30 / 50 |
| Optimizer | AdamW (lr $10^{-4}$, wd $10^{-3}$) |
| Batch size | 512 |
| Temperature $\tau$ | 0.07 |
| Hard negatives (training) | up to 3 state + 3 identity per instance |
| Evaluation pool | fixed 1 GT + 9 negatives; target composition is up to 3 state + up to 3 identity, with backfill across hard-negative pools and easy negatives used to fill remaining slots |
| Loss weights | $\lambda_s = 5.0$ (state), $\lambda_i = 1.0$ (identity) |
| Ensemble form | Full Ensemble: $S = A + w_v B + w_p C$ |

**Inference Details.** At inference time, we use the three score components and the Full Ensemble rule defined in Sec. 2.4. In the main CLIP-based experiments, these scores are computed in the frozen CLIP embedding space. For the universality results in Table 3, the same score definitions are applied analogously within each backbone's native frozen text-video embedding space: $A$ denotes the native text-video similarity, $B$ denotes the similarity between the last context clip and the candidate clip, and $C$ denotes the similarity between the CAST-predicted next-state embedding and the candidate clip. For the Full Ensemble score in Eq. (16), the ensemble coefficients are selected on a held-out validation split for each dataset/backbone setting using a fixed grid search with $w_v \in \{0.0, 0.1, \ldots, 0.5\}$ and $w_p \in \{0.2, 0.3, \ldots, 1.5\}$. Once selected, these coefficients are frozen and applied to all reported results within that setting. We do not tune coefficients per query, per example, or on the evaluation split.

### B.1. Additional Ablation: Context History Length

Figure 5 illustrates the impact of the context window length $L$ on model performance across YouCook2, COIN, and CrossTask. Across all three datasets, the largest gain occurs when moving from the context-agnostic setting ($L = 0$) to using only the immediate predecessor ($L = 1$). In particular, both Accuracy and Ident. Acc. improve sharply at $L = 1$, highlighting that even minimal visual history provides a strong cue for resolving procedural continuity.

As $L$ increases beyond 1, the gains largely saturate, with only modest additional improvements or small fluctuations across datasets and metrics. This suggests that the immediate predecessor $v_{t-1}$ serves as the primary causal anchor, while longer history provides only diminishing returns.

### B.2. Additional Retrieval Breakdown

## C. Appendix: Additional Qualitative Results and Evaluation Details

### C.1. Human Evaluation Protocol for Generation

To evaluate whether CAST improves candidate selection for black-box video generation, we conduct a blind human study on 300 prompts randomly sampled from the YouCook2 validation benchmark using a fixed random seed. Each prompt is

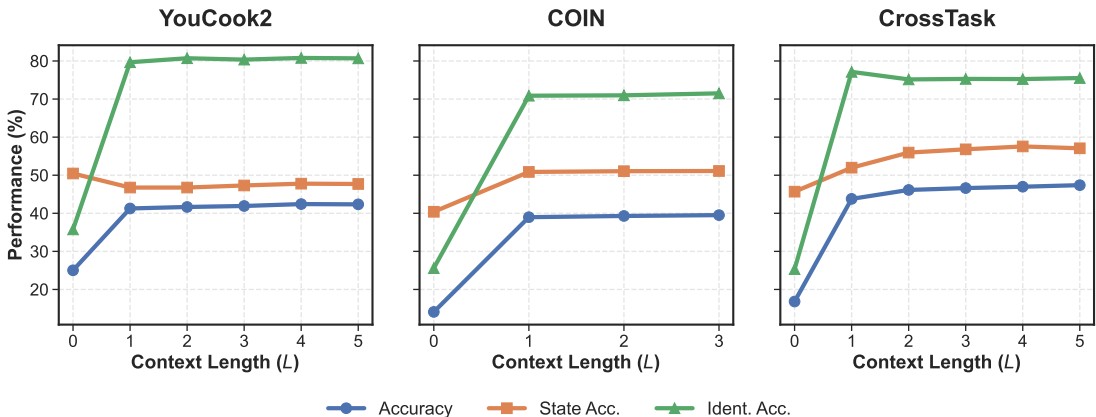

*Figure 5.* **Effect of context history length** ($L$). Results are shown on YouCook2, COIN, and CrossTask. Performance improves sharply from $L = 0$ to $L = 1$, and then largely saturates as $L$ increases further.

constructed from a benchmark query by pairing the query instruction text with a single visual context image. Specifically, we extract the context image as the last frame of the most recent clip in the query's context history.

For generation, we use Veo to produce $K = 4$ candidate continuation videos per prompt. The generation prompt follows a simple template: *"This image shows the context of the previous step. Generate a video for the next step: [instruction]."* We intentionally keep this prompt simple and rely on CAST at reranking time, rather than introducing additional handcrafted prompt constraints.

**Generation Setup and Reproducibility.** We use the Veo API to generate candidate continuation videos from the same context image and instruction prompt. For each prompt, we obtain $K = 4$ candidates from repeated generation calls using different random seeds $\{0, 1, 2, 3\}$. Unless otherwise specified, all other generation parameters follow the API defaults. We use the `veo-3.1-fast-generate-preview` model with 720p resolution, 16:9 aspect ratio, and 8-second duration. If a generation call fails or returns an invalid output, we retry until a valid output is obtained. In one case, generation repeatedly failed for a prompt despite multiple retries, likely due to the API safety filter. That prompt was discarded and replaced by a newly sampled prompt, so that the final human evaluation was conducted on 300 valid prompts. For reranking, we use the CLIP-B/32 CAST checkpoint trained on YouCook2, and encode each generated video using the same frame sampling and frozen video encoder as in retrieval.

We then apply two ranking strategies to the same candidate pool: (1) **Standard Text Match**, which ranks candidates using only the text-video matching score with the query instruction; and (2) **CAST Reranking**, which applies the same final reranking strategy as in retrieval, combining semantic matching, visual continuity, and CAST-predicted next-state compatibility. For each method, we retain the top-ranked candidate as its final selected output.

When the two methods select different candidates, annotators are shown the two selected videos in randomized order without method identities. We use three independent annotators for each prompt. Annotators compare the two videos under the following criteria:

- **Overall Preference**: which video is better overall as the continuation of the given context and instruction.

- **Physical Plausibility**: whether the object interactions, motion, and scene dynamics remain physically coherent.

- **Temporal Logic**: whether the generated continuation follows the procedural state implied by the visual context and instruction.

For each criterion, annotators select one of four options: *Video A*, *Video B*, *Tie*, or *Cannot judge*. We aggregate judgments using majority vote across the three annotators and report preference rates over all non-overlap prompts. If no majority exists (e.g., *A / B / Tie*), we record the outcome as a human tie. In the final study, no prompt was marked as *Cannot judge* after aggregation.

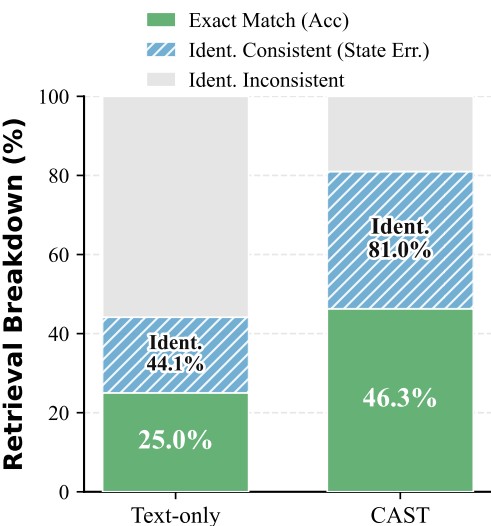

*Figure 6.* **Retrieval quality breakdown.** We categorize top-1 retrieval outcomes as *Exact Match* (green), *Identity Consistent but State-misaligned* (blue), or *Identity Inconsistent* (gray). CAST yields identity-consistent outcomes in 81.0% of queries (green+blue), substantially improving over the text-only baseline. This differs from the Ident. Acc. metric in Table 5, which requires the ground-truth clip to rank above all identity negatives.

When both ranking methods select the same candidate from the $K = 4$ generated videos, we record the case as an **overlap**. In our evaluation, this occurs for 142 out of 300 prompts (47.3%), leaving 158 non-overlap prompts (52.7%) for pairwise preference aggregation. Such cases are excluded from pairwise preference aggregation, since identical outputs admit no meaningful comparison. Equivalently, CAST changes the final selected candidate on 158/300 prompts (52.7%), defining the maximal subset on which pairwise human comparison is meaningful. We report the main results on this non-overlap subset in Table 6.

### C.2. Qualitative Analysis of Generation Reranking

We visualize a qualitative generation reranking example in Figure 7 and summarize the corresponding human evaluation protocol in Figure 8. Consider the query *"add ginger garlic paste"* following a visual context of frying onions. The baseline text score is often distracted by semantic overlap and consequently over-ranks candidates in which the ingredients already appear mixed or the procedural stage is mismatched, leading to **state errors**.

In contrast, **CAST** more reliably identifies the correct continuation by explicitly modeling the visual state transition $\Delta$. By extrapolating the current state $v_{t-1}$ under the query instruction, CAST captures the causal constraint induced by the action and favors candidates that match the expected next procedural state rather than scenes that merely contain semantically related objects.

Figure 8 illustrates the human evaluation setup used in Table 6. For each prompt, Veo generates $K = 4$ candidate videos from the same context frame and instruction. Standard text matching and CAST reranking then each select one top-ranked candidate from the same generated pool, and annotators compare the two selected outputs in a blind A/B setting.

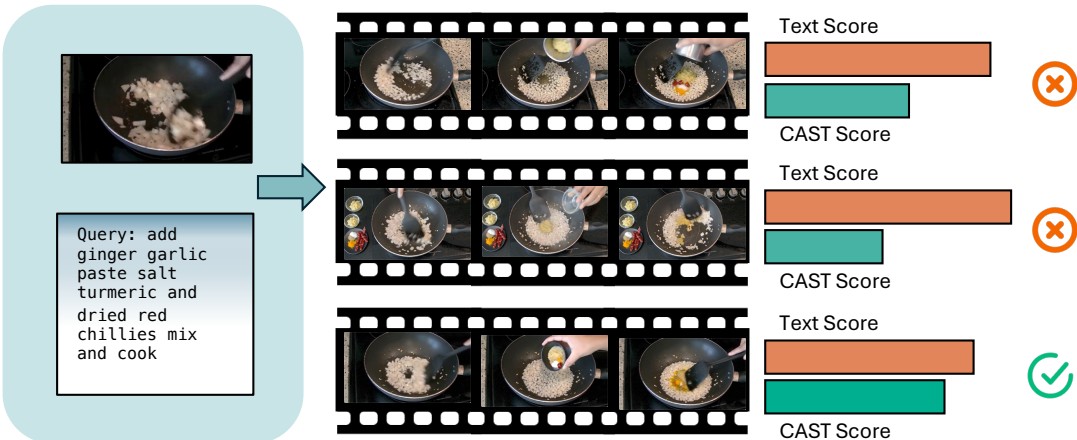

*Figure 7.* **Qualitative example of generation reranking.** Given a visual context and next-step instruction, text-only ranking is distracted by semantic similarity and consequently over-ranks procedurally mismatched candidates, whereas CAST more faithfully captures the expected next procedural state and selects a more coherent continuation.

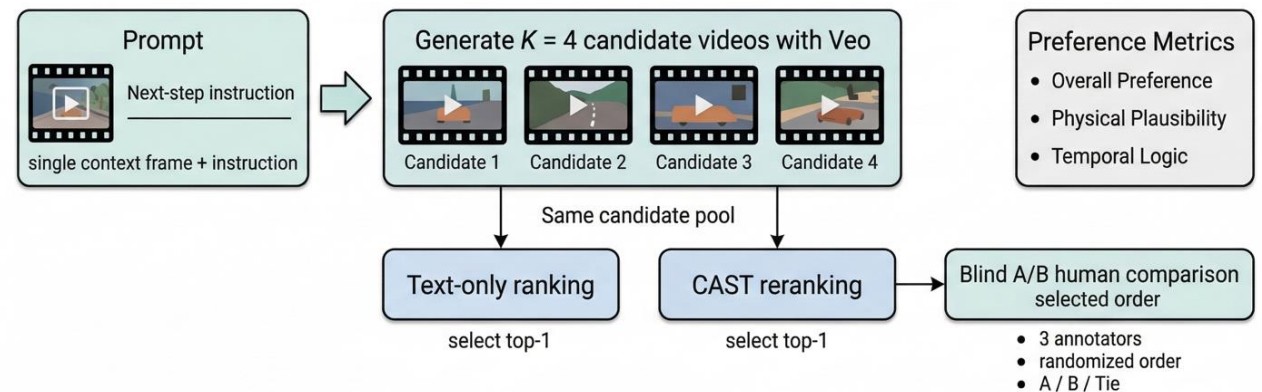

*Figure 8.* **Human evaluation protocol for generation reranking.** For each prompt, Veo generates $K = 4$ candidate videos from a single context frame and a next-step instruction. Standard text matching and CAST reranking each select one top-ranked output from the same candidate pool, and annotators then compare the two selected videos in a blind A/B setting using three annotators, randomized order, and the options *A*, *B*, *Tie*, or *Cannot judge*.

