# OpenReview forum: "CAST: Modeling Visual State Transitions for Consistent Video Retrieval"
_ICML.cc/2026/Conference — ICML 2026 regular_

### Official Review · Reviewer_XpfQ · 2026-03-09

**Soundness:** 3
**Presentation:** 3
**Significance:** 2
**Originality:** 2
**Overall Recommendation:** 4
**Confidence:** 3

**Summary:**

This paper addresses the consistency issue in video retrieval and proposes the Consistent Video Retrieval (CVR) task. It also designs a lightweight adapter module named CAST, which enhances the causal consistency and identity consistency of the retrieval by modeling state transitions.

**Compliance With Llm Reviewing Policy:**

Affirmed.

**Key Questions For Authors:**

1. For the generation reranking study, which generator(s) were used in practice, and how many candidates per prompt were considered? Are results consistent across domains?

2. Can CAST be extended to handle longer memory (beyond L=5)?

**Limitations:**

Please see the above.

**Strengths And Weaknesses:**

**Strength:**
1. The paper is well organized.

2. The topic of long video retrieval via modeling visual state transitions is worth exploring in the research community. This paper targets this important problem.

3. The paper provides many figures and related tables to help reviewers better understand of the proposed method.

**Weaknesses:**

There are some questions and concerns:

1.	Tables 1 and 2 mainly compare different fusion methods with each other and with themselves, but do not include comparisons with SOTA methods.

2.	Residual update as a mechanism for preserving identity is argued intuitively. Could the author devise more methods to verify the effectiveness of his/her approach?

3.	Figure 5 of the paper shows that the performance improvement is the greatest when L = 1. When L > 5, it reaches saturation. The performance when exploring longer time dependencies (such as L = 10) has not been investigated. It is suggested to supplement the long-context analysis or explain why it is not applicable to longer sequences.

4.	There is a lack of sensitivity analysis for the six hyperparameters in the public announcement.

**Please note that I am not an expert in this field, and please consider the reviews from others more.**

---

> ### Author Rebuttal · Authors · 2026-03-30
>
> We thank the reviewer for the constructive comments and questions.
>
> ### 1. Why Tables 1 and 2 do not compare against prior "SOTA methods"
> CVR is a new diagnostic task with a context-aware 1-vs-9 protocol and explicit state/identity hard negatives, so there are currently few prior methods evaluated under the same setting. For this reason, Tables 1 and 2 compare CAST against the most relevant adapted baselines under identical frozen backbones, including context-free retrieval, heuristic late fusion, learned late fusion, and early fusion. This controlled setup is intended to isolate whether explicit state-transition modeling improves over standard context aggregation under the same evaluation protocol. Table 2 further applies CAST on top of strong modern retrieval backbones, including InternVideo2, VideoPrism, GME-Qwen2-VL-2B, and Qwen3-VL-Embedding, to show that the gain is not tied to a particular encoder.
>
> ### 2. Verifying the residual update beyond intuition
> We agree that the identity-preservation role of the residual update should be supported empirically. Table 3 already provides such evidence: under the same early-fusion setting, residual prediction improves over direct target prediction (Acc 35.99 -> 38.95; State 38.92 -> 43.51), showing that modeling the continuation as an update from the anchor state is materially better than predicting a new target from scratch. Our YouCook2 error audit is consistent with this interpretation, suggesting that the residual formulation better preserves identity while improving state consistency, although the overall ceiling is still bounded by the frozen backbone.
>
> ### 3. Longer-memory analysis beyond L = 5
> We have extended the context-length analysis beyond the submitted Figure 5. Specifically, we constructed a fixed YouCook2 subset with at least 10 prior clips (175 queries), trained a max-context-10 CAST model, and evaluated L = 0, ..., 10 on the same subset. The trend remains consistent: CAST rises from 22.9 Acc at L = 0 to 57.1 at L = 1, and then remains in the 57.7-58.9 range through L = 10. The full ensemble rises from 30.3 at L = 0 to 60.0 at L = 1, peaks at 61.1 for L = 7-9, and remains 60.6 at L = 10. We therefore do not observe degradation up to L = 10; instead, performance saturates after the immediate predecessor. This supports our interpretation that short-horizon continuity is the dominant signal in the current procedural setting, while stronger long-range gains will likely require compressed or hierarchical memory rather than a larger flat window.
>
> ### 4. Generation reranking details
> In the reported generation experiment, we used Veo-3.1. For each prompt, we generated K = 4 candidate continuations from the same context frame and instruction, for 300 procedural prompts and 1200 candidates total. CAST then applied the same reranking used in retrieval, combining semantic matching, visual continuity, and predicted next-state compatibility, and selected the top candidate from each 4-way pool. The two selection methods agreed on 142 / 300 prompts and differed on 158 / 300 prompts; pairwise human preference aggregation was then conducted on the 158-prompt disagreement subset with three annotators. We will clarify that this is currently a procedural proof-of-concept rather than a cross-domain generator benchmark.
>
> ### 5. Hyperparameters
> We agree that sensitivity analysis beyond context length is useful. Appendix B reports the full training and inference configuration, which we kept fixed across datasets and backbones to avoid per-dataset retuning. Beyond the extended context study, we now ran additional checks. On the main YouCook2 validation set, for the semantic ensemble (text matching + CAST prediction), performance peaks at a CAST weight of 0.7 (45.53 Acc) and remains stable for nearby choices: 0.8 gives 45.46, 0.9 gives 45.35, 1.0 gives 45.10, and 1.1 gives 45.06. For the full ensemble (text + visual continuity + CAST prediction), the best setting is $(w_{vis}, w_{cast}) = (0.1, 0.6)$ with 46.0 Acc, while the adjacent $(0.1, 0.7)$ still reaches 45.7. On the fixed long-history subset, we further performed 3-point sweeps over the training hyperparameters, using the semantic ensemble and fixing all other settings. This gives Acc 55.7 / 56.2 / 50.5 over $\lambda_s = \{2.5, 5.0, 10.0\}$, 54.7 / 56.2 / 55.7 over $\lambda_i = \{0.5, 1.0, 2.0\}$, and 55.4 / 56.2 / 54.7 over $\tau = \{0.05, 0.07, 0.10\}$. While this is not an exhaustive sweep over all six hyperparameters, these checks suggest that CAST is reasonably stable around the default setting, whereas overly large $\lambda_s$ can over-emphasize state discrimination and hurt overall retrieval.

---

> > ### Author Rebuttal · Reviewer_XpfQ · 2026-04-02
> >
> > I have no other questions at the moment. I will wait for the responses from the other reviewers. Therefore, I will keep the original score unchanged.

---

### Official Review · Reviewer_Pjf7 · 2026-03-12

**Soundness:** 3
**Presentation:** 3
**Significance:** 2
**Originality:** 3
**Overall Recommendation:** 4
**Confidence:** 4

**Summary:**

This paper introduces the task of Consistent Video Retrieval (CVR), which focuses on retrieving video segments that are not only semantically relevant to a query but also temporally and procedurally consistent with the visual context. The authors argue that conventional video retrieval approaches prioritize local semantic similarity while ignoring state transitions and identity consistency across time, which often results in retrieval errors such as selecting clips where an action has already been completed or where the scene identity changes.

To address this issue, the paper proposes CAST (Context-Aware State Transition), a lightweight framework that models visual state transitions from the history of a video sequence. CAST learns a latent residual update representing procedural progression and uses it to retrieve clips that better match the expected next state of the process.

The authors also introduce a benchmark setup for the CVR task using existing instructional video datasets, including YouCook2, COIN, and CrossTask, and evaluate CAST against standard retrieval baselines. Experimental results show improvements in retrieval consistency across multiple datasets. Additionally, the authors demonstrate that the learned transition representation can be used to guide black-box video generation systems toward more temporally consistent outputs.

Overall, the paper addresses an interesting problem in video understanding and proposes a state-transition-based retrieval approach for improving temporal and identity consistency in video retrieval pipelines.

**Compliance With Llm Reviewing Policy:**

Affirmed.

**Final Justification:**

Thank you for the detailed and thorough rebuttal.

The additional clarifications regarding the CAST architecture and training objective significantly improve the clarity and reproducibility of the method. The explanation of how the transition representation is learned and integrated with the retrieval embedding space is particularly helpful.

I also appreciate the additional analyses on robustness, including cross-dataset transfer results and extended temporal horizon experiments. The saturation behavior with longer context windows and the detailed failure case analysis provide valuable insights into the strengths and limitations of the approach.

The pilot experiment on a less structured dataset (ActivityNet Captions) is a useful step toward understanding generalization beyond procedural videos, although the evidence remains somewhat limited.

Regarding novelty, I agree that the method is lightweight, but the combination of the CVR task formulation and the residual state-transition modeling provides a meaningful contribution.

Overall, the rebuttal addresses most of my concerns and strengthens both the empirical validation and clarity of the paper.

Updated score: Weak Accept

**Key Questions For Authors:**

1. The proposed method is evaluated mainly on instructional video datasets. How well do the authors expect the state transition modeling approach to generalize to more diverse video domains beyond procedural tasks?

2. Could the authors provide more details about the training objective used to learn the transition representation and how it interacts with the retrieval embedding space?

3. How sensitive is CAST to the length of the visual history used as context? For example, does performance degrade when longer context windows are used?

4. Have the authors analyzed common failure cases where CAST still retrieves inconsistent clips? Understanding these cases could help clarify the limitations of the approach.

**Limitations:**

The authors partially discuss the limitations of the proposed method, particularly its focus on procedural video datasets where state transitions are easier to model. However, the paper could further elaborate on potential limitations related to generalization across different video domains and the dependence on well-structured temporal progression in the data

**Strengths And Weaknesses:**

Strengths:


1) The paper is generally technically sound. The proposed formulation of consistent video retrieval and the modeling of state transitions using residual updates from visual history are reasonable design choices. The experimental evaluation includes multiple datasets and compares against relevant retrieval baselines, showing consistent improvements in retrieval performance and temporal consistency. The methodology appears appropriate for the problem setting.



2) The paper is mostly well organized and clearly written. The motivation for introducing consistent video retrieval is clearly explained, and the examples illustrating state errors and identity errors help clarify the problem setting. The figures and diagrams describing the CAST framework help the reader understand the overall pipeline.



3) The work addresses a relevant challenge in long-form video understanding and retrieval. As video generation and compositional video editing systems become more common, ensuring temporal consistency between clips becomes increasingly important. The proposed CVR task could potentially encourage future research in more context-aware video retrieval systems.



4) The idea of modeling state transitions from visual history for retrieval consistency is an interesting direction. The work also contributes by defining a new evaluation task (CVR) that explicitly measures temporal and identity consistency in retrieval systems.

...................................................................

Weaknesses:


1) Although the proposed approach shows improvements over baseline retrieval methods, the experimental analysis remains somewhat limited. The paper primarily evaluates retrieval accuracy and consistency but provides limited insight into how robust the learned transition representation is under different video domains or longer temporal horizons. Additional analysis of failure cases or ablation studies could strengthen the technical validation.



2) While the overall narrative is clear, some implementation details of the CAST architecture and training procedure are not fully specified. In particular, more details on how the transition representation is learned and how it interacts with the retrieval model would improve reproducibility.



3) The scope of the work is somewhat limited to instructional or procedural videos where state transitions are well defined. It is not entirely clear how well the proposed method would generalize to more diverse video domains such as open-world video retrieval or unconstrained video understanding.



4) Although the formulation of consistent video retrieval is interesting, the underlying method appears to be a relatively lightweight extension of existing retrieval pipelines with an additional transition modeling component. The overall methodological novelty is therefore moderate rather than highly substantial.

---

> ### Author Rebuttal · Authors · 2026-03-30
>
> We thank the reviewer for the feedback and address the main concerns below.
>
> ### 1. Scope beyond instructional / procedural videos
> We agree that our main evaluation is procedural by design: CVR requires measurable state progression, and instructional videos provide a controlled first setting. To probe beyond this regime, we built a weakly structured ActivityNet Captions pilot with an analogous sequence-level retrieval protocol, using the true next clip as the positive and semantically similar but temporally inconsistent clips as distractors, and ran 5 folds over 40 videos (196 clips). Text+CAST improves Acc from 55.52±4.62 to 66.52±6.86, MnR from 1.84±0.15 to 1.48±0.11, and pilot state consistency from 55.28±7.71 to 61.58±10.35. Gains are smaller and noisier than on instructional datasets, but Text+CAST improves both Acc and MnR in all 5 folds, suggesting that context-aware transition modeling remains a promising direction beyond tightly structured procedural videos.
>
> ### 2. CAST architecture, training objective, and interaction with retrieval embeddings
> CAST is a lightweight query-side adapter over frozen backbone embeddings. Given the last observed clip embedding $v_{t-1}$, recent visual history $H_t$, and current text query $q_t$, CAST predicts a residual transition $\Delta(v_{t-1}, q_t, H_t)$ in the same frozen space and forms $\hat{v_t} = \phi(v_{t-1} + \Delta)$, where $\phi$ is L2 normalization. The residual has an instruction-conditioned path and a history-attention path. During training, the backbone is frozen and only the adapter is optimized, using an in-batch contrastive term plus type-aware state-negative and identity-negative losses to align $\hat{v}_t$ with the true continuation while rejecting temporally or identity-inconsistent distractors. At inference, the gallery stays indexed once in the frozen space; CAST only updates the query, and its score can be combined with semantic and visual scores in the default full ensemble, so no gallery retraining or re-indexing is needed.
>
> ### 3. Robustness of the learned transition representation
> Within the scoped procedural setting, we evaluate robustness across three datasets with different procedural domains, multiple frozen backbones, and fusion baselines including heuristic late fusion, learned weighting, and early fusion, to separate explicit transition modeling from simple context aggregation. As an additional analysis, under procedural domain shift, CAST trained on YouCook2 / COIN improves CrossTask (Acc +12.11 / +12.65; MnR -1.35 / -1.31), suggesting that the learned transition signal is not tightly tied to a single dataset.
>
> ### 4. Longer temporal horizons
> We agree that longer temporal reasoning is important. The submitted paper already studies context length in Fig. 5 for $L = 0, ..., 5$. To test longer horizons, we rebuilt a fixed YouCook2 subset with at least 8 prior clips and evaluated 404 queries while sweeping $L = 0, ..., 8$ on the same samples. CAST jumps from 24.5 Acc at $L=0$ to 54.2 at $L=1$, then remains stable through $L=8$ (53.5-55.0 Acc; MnR 1.94-1.95). The full ensemble shows the same pattern (28.5 to 56.2 at $L=1$, then 54.2-56.2 through $L=8$). This supports saturation rather than clear degradation: the immediate predecessor is the strongest anchor, while extending the window beyond that brings limited additional benefit.
>
> ### 5. Failure cases and limitations
> We agree that stronger failure analysis is useful. On YouCook2, the full ensemble fixes far more text-only errors than it introduces (743 fixes vs. 163 regressions). In particular, it fixes 156 cases where the text-only top-1 error is a state-negative and 498 where it is an identity-negative. The remaining and newly introduced errors are concentrated on hard state negatives: among the 163 cases where the text-only baseline is correct but the full ensemble fails, 127 retrieve a state-negative as top-1, versus 31 identity negatives. This pattern is consistent with CAST leveraging transition cues, but sometimes over-preferring a state-consistent yet incorrect continuation. Failures also arise when the needed cue is weakly preserved in pooled frozen embeddings, such as subtle local state changes, longer-range dependencies, or fine-grained identity distinctions.
>
> ### 6. On methodological novelty
> We agree that CAST is lightweight by design and do not claim novelty through architectural complexity alone. Instead, the key novelty lies in the formulation and benchmark: CAST introduces a residual state-transition view of retrieval, modeling the next-step representation as an update from the anchor state rather than generic context aggregation, while CVR makes state/identity-consistency failures measurable under a shared protocol in a way standard retrieval metrics miss. Together, this provides a targeted framework for context-aware retrieval. We will revise the framing accordingly without overstating architectural complexity.

---

### Official Review · Reviewer_Wafo · 2026-03-13

**Soundness:** 4
**Presentation:** 4
**Significance:** 4
**Originality:** 3
**Overall Recommendation:** 5
**Confidence:** 4

**Summary:**

This paper introduces the novel task of Consistent Video Retrieval (CVR) to address the limitations of standard context-agnostic retrieval systems, which often return semantically relevant but temporally or physically incoherent clips (e.g., state or identity errors). To tackle this, the authors propose CAST (Context-Aware State Transition), a lightweight, plug-and-play adapter that operates on top of frozen vision models. Instead of directly matching text to video, CAST predicts a state-conditioned residual update ($\Delta$) based on the anchor state, visual history, and the text query, reformulating the target as $\hat{v}_{t}=\phi(v_{t-1}+\Delta(v_{t-1},q_{t},\mathcal{H}_{t}))$. Evaluated on a newly constructed benchmark with hard state and identity negatives across YouCook2, COIN, and CrossTask, CAST significantly outperforms existing aggregation baselines. Furthermore, it demonstrates practical utility as a consistency verifier for downstream black-box video generation models.

**Compliance With Llm Reviewing Policy:**

Affirmed.

**Key Questions For Authors:**

1. How robust is the additive residual transition mechanism when handling consecutive procedural steps separated by severe camera cuts or dramatic viewpoint changes? Does the performance degrade when visual inertia is completely broken?
2. To overcome the fixed context length limitation ($L=5$), have you considered incorporating recurrent memory mechanisms or global memory tokens to compress and retain long-term historical context?

**Limitations:**

See above

**Strengths And Weaknesses:**

Strengths
1. The identification of "state" and "identity" errors in procedural video retrieval is highly insightful. The proposed CVR benchmark, featuring specifically mined hard negatives, provides a robust testing ground for temporal and causal consistency.
2.  Modeling procedural actions as residual state transitions in the latent space is an elegant design. The additive update naturally preserves static identity features while selectively modifying the task-relevant state, avoiding complex gating or tracking mechanisms.
3. CAST is embedding-agnostic and acts as a lightweight adapter. Its demonstrated effectiveness across various foundation models (CLIP, InternVideo2, VideoPrism, Qwen-VL) and its application in guiding video generation highlight its immense practical value and deployment potential.

Weaknesses
1. As noted by the authors, CAST relies on a fixed, short-term context window ($L=5$). This restricts its ability to perform long-term causal reasoning, such as referencing a tool or object prepared several minutes prior in a video.
2. The core assumption of an additive residual transition inherently assumes a degree of visual continuity. It is unclear how effectively this linear shift handles abrupt camera cuts or drastic viewpoint changes where the entire visual identity matrix of the scene transforms.
3. Because CAST is a lightweight adapter on frozen encoders, it is fundamentally limited by the base model's discriminative power. If the backbone cannot extract fine-grained state differences (e.g., subtle texture changes before and after cooking), CAST cannot correct for this lack of information.

---

> ### Author Rebuttal · Authors · 2026-03-30
>
> We thank the reviewer for the careful reading and for highlighting both the task value and the practical utility of CAST.
>
> ### 1. Robustness to severe cuts or viewpoint changes
> We agree that the additive residual formulation assumes some local continuity, and abrupt cuts or strong viewpoint shifts are a meaningful stress test. To probe this directly, we ran an additional analysis on the YouCook2 benchmark by stratifying queries by anchor-to-ground-truth cosine similarity in the frozen clip embedding space, using low similarity only as a coarse proxy for disrupted short-term visual continuity rather than a literal camera-cut annotation. In the lowest-similarity quartile (692 queries, cosine <= 0.827), the visual-continuity baseline collapses to 3.5 Acc / 5.6 State, while text-only remains at 27.6 Acc. In the same subset, CAST still reaches 36.3 Acc, and the full ensemble reaches 38.2 Acc. By contrast, in the highest-similarity quartile, the visual baseline is much stronger (49.2 Acc), and the full ensemble reaches 52.5 Acc. We observe the same qualitative pattern under the same stratification procedure with a stronger VideoPrism backbone: within its own lowest-similarity quartile, the visual baseline drops to 3.5 Acc, while CAST and the full ensemble remain at 69.1 and 71.0 Acc, respectively. While this proxy is admittedly coarse and may also capture harder state changes, CAST remains clearly stronger than the visual baseline under this stress test, indicating that the transition predictor contributes beyond simply copying the previous visual state.
>
> ### 2. Fixed context length and longer-term memory
> We agree that recurrent memory or global memory tokens are a natural next extension. The submitted benchmark caps the available history at $L = 5$, so the main paper focused on short-horizon consistency. In an additional extended-context analysis on a fixed YouCook2 subset with at least 8 prior clips, we evaluated $L = 0, ..., 8$. CAST and the full ensemble both show a large jump from $L = 0$ to $L = 1$, followed by saturation rather than clear degradation up to $L = 8$. Our reading is that the immediate predecessor is the strongest anchor in the current procedural setting, while truly long-range dependencies will likely require compressed or hierarchical memory rather than a larger flat window.
>
> ### 3. Scope of the current adapter
> We also agree with the reviewer that CAST cannot recover information missing from the frozen backbone. We view this as a deliberate first step: the paper isolates the value of explicit transition modeling on top of strong off-the-shelf embeddings, and a natural next direction is to combine the CVR formulation with token-level or region-level video representations that preserve finer object-state and instance-level evidence.

---

### Official Review · Reviewer_V1rp · 2026-03-15

**Soundness:** 3
**Presentation:** 3
**Significance:** 3
**Originality:** 3
**Overall Recommendation:** 4
**Confidence:** 3

**Summary:**

This paper introduces a new task, Consistent Video Retrieval (CVR). To address this problem, the paper proposes CAST, a lightweight adapter built on top of frozen video or multimodal backbones. CAST models the next-step visual representation as a residual state transition conditioned on the recent visual history and the current instruction, using a context encoder with cross-attention and a contrastive training objective. The authors further construct a CVR benchmark based on YouCook2, COIN, and CrossTask. Experiments show that CAST improves retrieval performance over standard text-based baselines and several fusion-based context baselines, and can also be used to rerank generated video candidates to improve temporal logic.

**Compliance With Llm Reviewing Policy:**

Affirmed.

**Key Questions For Authors:**

1. Could the authors provide stronger ablations to verify that the gains mainly come from explicit state transition modeling, rather than from hard negatives, loss design, or score ensembling?

2. Since CAST appears more consistently effective on state consistency than on identity consistency, could the authors provide a clearer analysis of where identity modeling still fails?

**Limitations:**

See above.

**Strengths And Weaknesses:**

**Strengths**

1. The paper formulates a well-motivated and practically meaningful problem. CVR explicitly emphasizes temporal coherence and identity consistency. The task motivation is easy to understand and clearly exposes an important limitation of current retrieval formulations.

2. The proposed method is conceptually simple and well aligned with the problem. Modeling the next clip representation as a residual transition is an intuitive design choice. As a lightweight adapter rather than a full model replacement, CAST is also easy to integrate with existing backbone features.

3. The experimental results support the main empirical claims. CAST shows large gains over context-free retrieval baselines and improves performance across multiple datasets and multiple frozen backbones.

**Weaknesses**

1. The technical novelty at the modeling level is somewhat limited. CAST is built from fairly standard components, and the main novelty lies more in the task framing and the way these ideas are combined.

2. The method appears strongly dependent on the quality of the backbone representations, which may limit its ceiling. Since CAST operates in a frozen clip-level embedding space, it cannot recover identity cues, object-instance information, or subtle visual states if these are not already encoded by the backbone. In this sense, CAST mainly reorganizes and amplifies consistency signals already present in the representation rather than solving the underlying representation bottleneck.

3. The use of a single global clip embedding may be too coarse for the target problem. Identity consistency and fine-grained state changes often depend on localized spatiotemporal details that may be washed out in a single pooled vector. As a result, the method may face intrinsic difficulty in handling instance-level identity continuity or subtle state distinctions.

---

> ### Author Rebuttal · Authors · 2026-03-30
>
> We thank the reviewer for the thoughtful comments and for highlighting both the task value and the lightweight design. CAST is intentionally lightweight at the architectural level: our goal is not a full model replacement, but a targeted adapter plus a benchmark that makes temporal, state, and identity consistency measurable.
>
> ### 1. Are the gains mainly from explicit transition modeling?
> In the paper, we compare against heuristic late fusion, learned weighting, and early fusion baselines to separate explicit transition modeling from simpler forms of context aggregation. The late-fusion baselines only re-rank candidates in the frozen embedding space, while the early-fusion baseline uses context to predict a target representation without CAST's structured dual-path residual transition.
>
> This becomes even clearer in an additional loss ablation on a fixed extended YouCook2 long-history subset. We construct this subset using the same rule as in our horizon analysis, namely requiring at least 8 prior clips for every query. On this subset, text-only retrieval obtains 25.2 top-1 Acc. Training the transition predictor without the hard-negative state/identity losses and evaluating CAST alone already raises this to 50.0, indicating that the main gain comes from explicit next-state prediction rather than from hard-negative mining or score ensembling. Adding the same full-ensemble inference used in the paper (text + visual continuity + CAST score) improves this no-hard-negative variant to 55.7. Restoring the hard-negative losses during training and evaluating with that same full ensemble yields 56.2 and a further improvement in state consistency (60.9 to 68.1). Overall, the dominant gain comes from explicit next-state prediction; the hard-negative terms sharpen the consistency signal, especially for state discrimination, but do not account for the effect by themselves.
>
> ### 2. Where identity modeling still fails
> Identity consistency improves substantially overall in the full ensemble: it fixes 498 text-only identity-negative errors, and among the 163 cases where text-only is correct but the full ensemble fails, only 31 become identity negatives, versus 127 state negatives. To further localize the remaining identity failures, we stratified queries by a frozen-space identity-margin proxy, defined as $\cos(\text{anchor}, \text{GT}) - \max_j \cos(\text{anchor}, \text{identity-neg}_j)$. The trend is clear: full-ensemble identity accuracy rises from 53.9 in the lowest-margin quartile to 71.3, 81.3, and 88.4 in the higher quartiles, while the lowest-margin quartile alone accounts for 177 / 450 = 39.3% of the remaining full-ensemble identity-negative errors on YouCook2. More broadly, the lowest two margin quartiles account for 50% of samples but 300 / 450 = 66.7% of those remaining identity-negative errors. This is consistent with the reviewer's observation: CAST improves identity consistency when the relevant cues are already separable in the frozen representation, but it does not recover missing local instance evidence from a single pooled clip embedding.
>
> ### 3. Limitation and intended scope
> We agree that frozen clip-level embeddings impose a ceiling on what CAST can recover. This is why we present CAST as a lightweight first step: it tests whether explicit state-transition modeling helps on top of existing embeddings, rather than claiming to solve the full representation bottleneck. A natural next direction is to combine the CVR formulation with stronger token-level or region-level video representations that preserve finer identity and object-state evidence.

---

### Decision · Program_Chairs · 2026-04-30

**Decision:**

Accept (regular)

**Comment:**

This paper introduces a new task called Consistent Video Retrieval (CVR) to evaluate models on "wrong state" or "wrong identity" errors. To solve this, the authors propose Context-Aware State Transition (CAST), a lightweight, plug-and-play adapter that models state transitions on top of frozen vision models.

All reviewers recommended acceptance (1 Accept, 3 Weak Accepts). Reviewers praised the clear motivation, the practical relevance of the CVR task, and the elegant design of the CAST adapter, which effectively preserves static identity features while updating task-relevant states. Reviewers initially questioned the method's reliance on short-term context and its generalization beyond procedural videos.

During the rebuttal phase, the authors provided extensive new experiments, including tests on extended temporal horizons (up to L=10) and a pilot study demonstrating effectiveness on non-procedural videos (ActivityNet). The reviewers found these additions convincing and maintained their positive scores, agreeing that the reliance on frozen backbones is a reasonable design choice for a lightweight adapter.

The paper is technically sound, well-motivated, and introduces a useful framework for the community. Therefore, we recommend acceptance.